# Modeling and Optimization of Surface Integrity and Sliding Wear Resistance of Diamond-Burnished Holes in Austenitic Stainless Steel Cylinder Lines

**Galya Duncheva** [1,*] , **Jordan Maximov** [1] , **Angel Anchev** [1] , **Vladimir Dunchev** [1] , **Yaroslav Argirov** [2] and **Svetlozar Velkov** [1]

1    Department of Material Science and Mechanics of Materials, Technical University of Gabrovo, 5300 Gabrovo, Bulgaria; jordanmaximov@gmail.com (J.M.); anchev@tugab.bg (A.A.); v.dunchev@tugab.bg (V.D.); s_velkov97@abv.bg (S.V.)

2    Department of Material Sciences, Technical University of Varna, 9010 Varna, Bulgaria; jaroslav.1955@abv.bg

*    Correspondence: duncheva@tugab.bg

**Abstract:** This article outlines a technology for hole-finishing in short-length cylinder lines to improve wear resistance. The technology is based on an optimized diamond-burnishing (DB) process. The latter was implemented on conventional and CNC lathes, milling machines, and machining centers using a simple burnishing device with an elastic beam. The material used in this study was AISI 321 austenitic stainless steel. The governing factors used were the radius of the diamond insert, burnishing force and feed rate. The objective functions relating to surface integrity characteristics were selected on the basis of their functional importance relative to the wear resistance of the processed hole surface: height and shape roughness parameters, surface microhardness, and surface residual axial and hoop stresses. The one-factor-at-a-time method (used to reduce the factor space), a planned experiment, and regression analyses were used. The multi-objective optimization tasks, which were defined for three diamond insert radius values of 2, 3, and 4 mm, were solved via the Pareto-optimal solutions approach available for a non-dominated sorting genetic algorithm (NSGA-II). Using the optimal values of the governing factors selected from the Pareto fronts, cylinder lines were processed. Samples were then cut from these cylinder lines for reciprocating sliding wear tests under two modes: dry friction and boundary lubrication friction. Additionally, wear test samples were cut from the cylinder line, which was finished with traditional grinding. A finite element simulation was then used to select an appropriate pressing force. The results obtained from the reciprocating sliding wear tests under both the dry and boundary lubrication friction regimes show that to minimize the wear on cylinder lines made of AISI 321 steel, DB with a diamond insert of radius 2 mm is the optimal finishing process.

**Keywords:** austenitic stainless steel; cylinder lines; diamond-burnishing; surface integrity; optimization; sliding wear

## 1. Introduction

AISI 321 is a high-grade titanium-stabilized chromium–nickel austenitic stainless steel (SS). The addition of titanium and niobium, with carbon activity significantly greater than that of the chromium, stabilizes this steel against the formation of chromium carbide, significantly increasing its resistance to inter-granular corrosion in the temperature range 450–850 °C in contrast to lower-grade 304 SS. In addition, 321 SS has high resistance even down to cryogenic temperatures, excellent resistance to stress corrosion cracking, and is non-magnetic. Due to its properties, AISI 321 SS is a preferred material for various industrial applications in a wide temperature range. For instance, it is used in aircraft components, chemical processing, and food equipment, and it is also used in medicine and the pharmaceutical industry. These applications often use cylinder-type components with

life cycles that are limited by the complex conditions of the surface layers around their holes. These areas are subjected to reversible sliding, usually in the presence of oil or an active medium (organic or inorganic substance), as a result of contact with other elements (most often a piston). Under these conditions, sliding wear under boundary lubrication is a crucial consideration.

To maximize wear resistance, an appropriate finishing method must be used to ensure the required surface integrity (SI) of the hole. The development of a suitable finishing process based on the correlation between SI and the desired operational behavior (including high wear resistance) is a major task falling within the scope of the surface engineering discipline [1]. Long holes are usually finished via honing due to the beneficial effect of the resulting cross-hatched cylinder liner texture on their tribological behavior [2,3]. Such a surface texture is particularly suitable when there is boundary lubrication friction. Grinding is widely used for finishing short-length holes.

However, the austenitic structure of AISI 321 results in low hardness and hence poor wear resistance. Two main approaches are effective for modifying the working surfaces of components made of austenitic chrome–nickel steels: cold working the surface and low-temperature thermo-chemical treatments (nitriding and carbonizing). When the goal is to modify the surfaces of short-length cylindrical holes, static burnishing processes are more suitable. For a finishing process based on a burnishing technique, roller burnishing and deep rolling processes [4] are preferred. They drastically reduce the height parameters for roughness, increase the micro-hardness, and introduce residual compressive stresses, which increase the wear and crack resistance of the hole surface. Finishing a hole via surface plastic deformation with rolling friction contact requires more complex and expensive tools (for instance, a hydrostatic sphere [4]) compared to those required for diamond-burnishing (DB) [5].

Regardless of the wide application of grinding as a finishing method, the resulting surface's physical–mechanical condition corresponds to that of a surface obtained via cutting. In addition, after grinding, a hole's micro-hardness is relatively low due to the absence of the cold work effect. In the present study, a DB-based technology is proposed to modify the SI of short-length holes in cylinders made of 321 SS.

Despite the wide application of burnishing technologies, research on hole burnishing is very scarce. According to [5], only 6% of the research on slide burnishing (including DB) is devoted to holes. Akkurt et al. [6] compared the effects of different finishing processes on the height parameters for the roughness of a hole in a cooper-based alloy and found that roller burnishing had a significant advantage: a significant reduction in these parameters and a significant increase in surface micro-hardness. Przybylski [7] used DB to finish the hole of a satellite of planet gear made of NiCrMo13 hardened steel and achieved a roughness of $R_a = 0.02$ μm. A method for obtaining regular depressions in the hole surface of a bearing sleeve via eccentric burnishing was developed by Korzynski [8]. Maximov et al. [9] used DB to increase the crack resistance of joint bar holes, whereas Pa [10] developed a combined finishing method for holes based on slide burnishing and electrochemical finishing. Finally, the effects of DB on the SI and sliding wear behavior of aluminum bronze holes were investigated by Duncheva et al. [11]. However, information is lacking on the effect of DB on the SI of chromium–nickel austenitic SS holes.

The effects of DB on the holes in cylindrical components can be maximized via the multi-objective optimization of the process parameters where the vector optimization criterion consists of suitably selected SI features of the hole surface. Therefore, the selected characteristics of SI must be maximally function-relevant in terms of surface contact in the presence of oil or another active medium, friction, and wear. For a given material, two sets of SI characteristics are of primary importance: geometric (surface texture parameters) and physical–mechanical characteristics. Currently, there are a large number of 2D and 3D parameters to consider for the surface texture that are classified into the following groups: height, spatial, hybrid, functional, and other [12,13]. Some parameters describe similar surface properties, while other parameters are correlated. Therefore, in addition to being

function-relevant, the selected roughness parameters should be limited in number, statistically independent, and have low sensitivity to measurement errors. The measurement of a 2D parameter is simpler and requires less time; therefore, 2D parameters were used in the present study. A number of studies were devoted to the functional significance of the height (amplitude) parameters of the surface texture for the tribological behavior of cylinder liners. The 2D parameters $R_{3z}$ and $R_{tm}$ have been used to characterize roughness profiles [14,15], while $R_p(S_p)$ and $R_v(S_v)$ have been used as wear indicators [16–18]. The behavior observed under different friction regimes depends on the magnitudes of the height parameters. Under fluid friction, a cylinder with a smoothly polished surface is exposed to seizure and even adhesion [19,20] due to the more difficult retention of oil, while a rougher SS hole surface increases seizure resistance [21]. In general, smooth surfaces are prone to seizure, while rough surfaces lead to more intense friction and wear [13]. The 2D (3D) height parameters $R_a(S_a)$ and $R_q(S_q)$ provide an integral picture of the roughness profile (surface). Minimizing these parameters reduces friction and decreases wear. The arithmetic mean deviation $R_a$ is used most often in the production process, as it provides a very good general description of height variations; however, it is not sensitive to small changes in the profile [22]. The root mean square deviation (RMS) $R_q(S_q)$ of the profile (surface) heights is more sensitive to the deviations than $R_a(S_a)$, and according to some authors [23], it is one of the parameters of determining functional importance for crack resistance under dynamic loads: a lower RMS value corresponds to a lower tendency towards fatigue. At the same time, deep valleys are more function-relevant in terms of fatigue behavior than peaks. A larger value of $R_v(S_v)$ is an indicator of local surface stress concentrators, making this parameter relevant for crack detection [13]. Although $R_a(S_a)$ and $R_q(S_q)$ have similar variation trends, there is no known relationship between their relative variation magnitudes [13]. Therefore, the height parameters $R_a$, $R_q$, and $R_v$ are important geometric indicators for the tribological behavior and crack resistance of a hole surface. The $R_k$ family functional parameters, which are related to the material ratio curve, provide information concerning the influence of surface amplitudes on friction and wear [13]. In particular, $R_{pk}$, $R_k$, and $R_{vk}$ correspond to the heights of the peak, core, and valley sections of the surface profile. It has been assumed that $R_{pk}$ is responsible for the running-in stage, $R_k$ is responsible for the steady state, and $R_{vk}$ is associated with oil retention. However, these assumptions have not been proven experimentally [13].

The functional significance of the shape parameters $R_{sk}(S_{sk})$ (skewness) and $R_{ku}(S_{ku})$ (kurtosis) for friction and wear is known [22]. A surface texture characterized by deep valleys and sharp peaks ($R_{sk}(S_{sk}) < 0$ and $R_{ku}(S_{ku}) > 3$) and relatively low values of the parameters $R_a(S_a)$ and $R_q(S_q)$ significantly improves lubrication under boundary lubrication friction conditions [11,22,24]. The importance of negative $R_{sk}(S_{sk})$ values for improving contact on rough surfaces [25–27], as well as for reducing friction and wear on smooth surfaces under dry and lubrication sliding conditions [11,28,29], has been confirmed. Shape parameters have been found to be sensitive to hydrostatic ball burnishing parameters [30]. Given that textured surfaces are skewed, this finding confirms the effect of surface texturing due to static burnishing processes. In general, improving the tribological behavior of sliding components through appropriate surface texturing has considerable potential. Taking into considering the sliding friction contact, it can be assumed that this effect is more pronounced after DB [11] than after static burnishing processes employing rolling friction contact. The orientation of an anisotropic one-directional surface to the sliding direction is also of significant importance to tribological behavior. The transverse orientation of the asperities typically leads to lower friction under the boundary lubrication friction condition than their longitudinal orientation [31,32]. DB kinematics and a low feed rate produce a sliding direction with an almost transversely oriented texture. Thus, it is of interest to evaluate the effect of the DB process on the sliding wear resistance of the hole surface. To evaluate the cold work effect and beneficial compressive residual stresses inherent in burnishing processes, particularly DB, it is appropriate to measure surface micro-hardness and surface hoop and axial residual stresses. In order to select

an appropriate combination of the governing factor magnitudes for the DB of holes in AISI 321 SS, a multi-objective optimization was carried out in the present study. Based on the above rationale, the vector optimization criterion was composed of the following SI characteristics: (1) the geometric characteristics $R_a$, $R_q$, $R_v$, $R_{sk}$, and $R_{ku}$ (2D roughness parameters); (2) the physical–mechanical characteristics of surface micro-hardness HV, plus the surface axial ($\sigma_a^{res}$) and hoop ($\sigma_t^{res}$) residual stresses.

In this article, a technology for processing holes in cylinder lines made of AISI 321 SS is presented. The technology, which is based on DB, was developed to improve the sliding wear on these cylinders. First, experimental research and the multi-objective optimization of the DB process were carried out. Second, an experimental study of the tribological behavior of samples processed under the optimized DB process was conducted. Finally, a comparison of the sliding wear behavior of these samples was made with that of samples processed by grinding. The comparison proves the advantage of the developed technology. Figure 1 shows the flow chart of the present study, which was conducted using seven main steps.

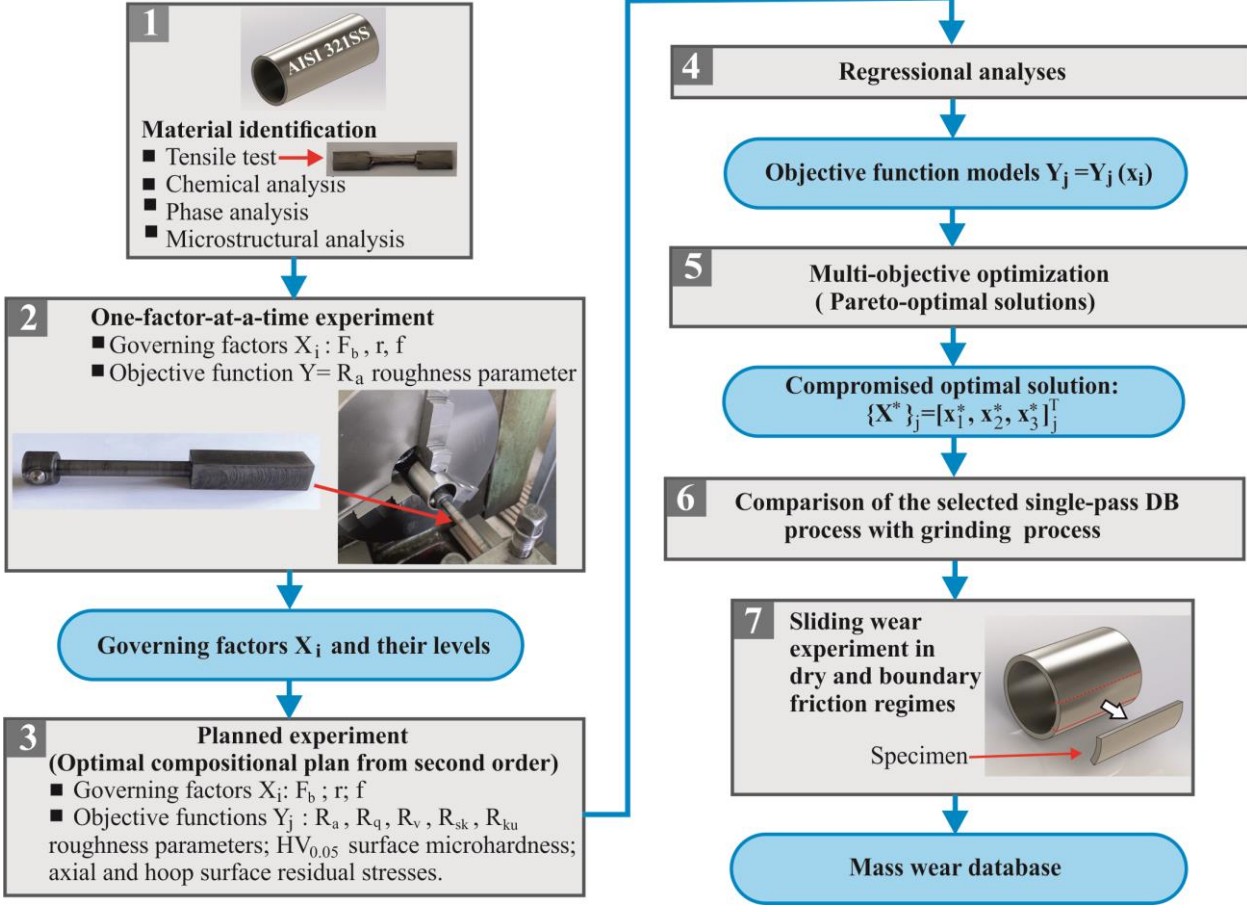

**Figure 1.** Flow chart of the present study.

## 2. Materials and Methods

A pipe of chromium–nickel AISI 321 austenitic SS with an outer diameter of 42 mm and a wall thickness of 4 mm was obtained. An optical emission spectrometer was then used to determine its chemical composition in weight percentages. The tube's basic mechanical characteristics were established at room temperature via tensile tests conducted on a Zwick/Roell Vibrophore 100 testing machine, where all values were determined as arithmetic means of three test results. Specimens were cut from the tube along its axis (Figure 2). Because the specimens' centers of gravity for their working-part cross-sections did not lie in the tensile load direction, their loadings were, in fact, eccentric

tension (superposition of tension and constant bending). However, due to the negligible eccentricity, it was assumed that tests were for pure tension. The hardness was measured via a ZWICK/Indentec–ZHVμ-S tester using a spherical-ended indenter with a diameter of 2.5 mm, loading of 63 kg, and holding time of 10 s. A Bruker D8 Advance X-ray Diffractometer was used to conduct the phase analysis. The peak positions were determined by the Crystallography Open Database. The strain-induced $\alpha'$-martensite contents in the surface layers were determined with Bruker's DIFFRAC.DQuant V1.5 specialized software [33]. The microstructures in the cross-sectional areas of the pipe were observed via optical microscopy (OM, NEOPHOT 2) with "Aqua regia" ($HNO_3$:HCl = 1:3).

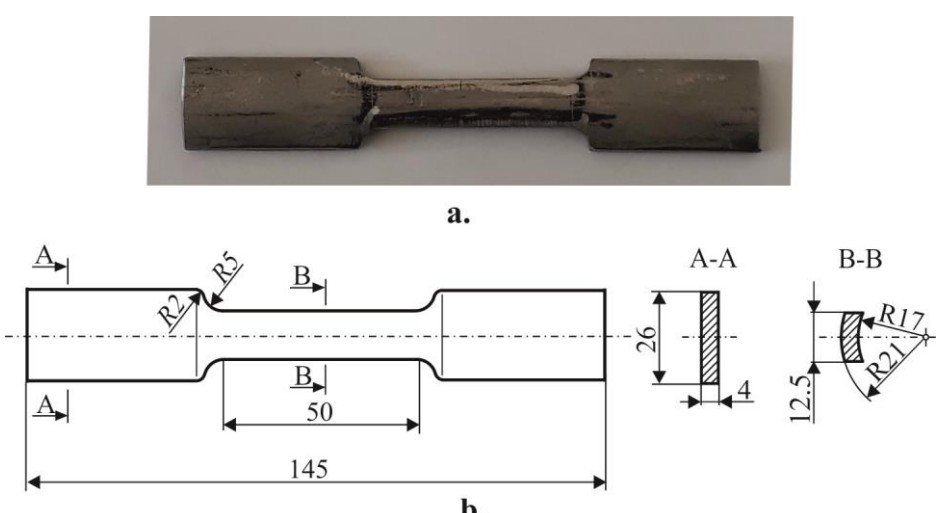

**Figure 2.** Shape and dimensions of tensile test specimens: (**a**) photo and (**b**) drawing.

DB was implemented on a C11 conventional lathe equipped with a specially developed device featuring an elastic beam and spherical-ended polycrystalline diamond insert (see Figure 1, step 2). The governing factors were the sphere radius r, burnishing force $F_b$, and feed rate f. The burnishing velocity v was 80 m/min. Hacut 795-H lubricant was used for the single-pass DB process. Consequently, the one-factor-at-a-time method and planned experiment were applied.

The roughness parameters were measured via a Mitutoyo Surftest SJ-210 surface roughness tester. The final values of the roughness parameters were the arithmetic averages obtained from the measurements on six equally spaced generatrixes.

A ZHVμ Zwick/Roell micro-hardness tester with a 10 s holding time and 0.05 kgf load was used to measure surface micro-hardness. Ten measurements were carried out for each specimen. The group median was then chosen as the final surface micro-hardness magnitude for each corresponding sample.

The residual surface stresses were measured using the $\sin^2 \psi$ method and a least-squares fitting procedure in conjunction with a Bruker D8 Advance X-ray Diffractometer. The effective penetration depth of the CrK$\alpha$ radiation was approximately $6 - 7$ μm.

Multi-objective optimization was conducted using the NGSA-II non-dominated sorting genetic algorithm [34,35].

The sliding wear resistance of a treated surface (as a pair with a spherical counter-body of diameter 10 mm and made of hardened bearing steel) was assessed via the reciprocating sliding wear test. The tribological tests were conducted under the same conditions for all specimens across two friction regimes: (1) dry friction and (2) boundary lubrication under which the oil was supplied at a flow rate of 2 drops per minute. The samples were cut from the corresponding cylinders, as shown in Figure 1, step 7. The specimen sizes are shown in Figure 3. The mass wear on each specimen for a given friction path was determined using the following methodology:

- Before friction, the specimen's initial mass $m_0$ was measured with an accuracy of 0.1 mg by means of a WPS 180/C/2 electronic balance. To prevent electrostatic effects, each specimen was cleaned with ethyl alcohol to remove mechanical and organic particles.
- The specimen mass $m_i$ was measured for a set friction path $L_i$. The mass wear $\Delta m_i$ was calculated with the formula: $\Delta m_i = m_o - m_i$, mg.
- The average wear rate $\gamma_i$ was calculated with the formula $\gamma_i = \frac{m_i}{L_i}$, mg/m.

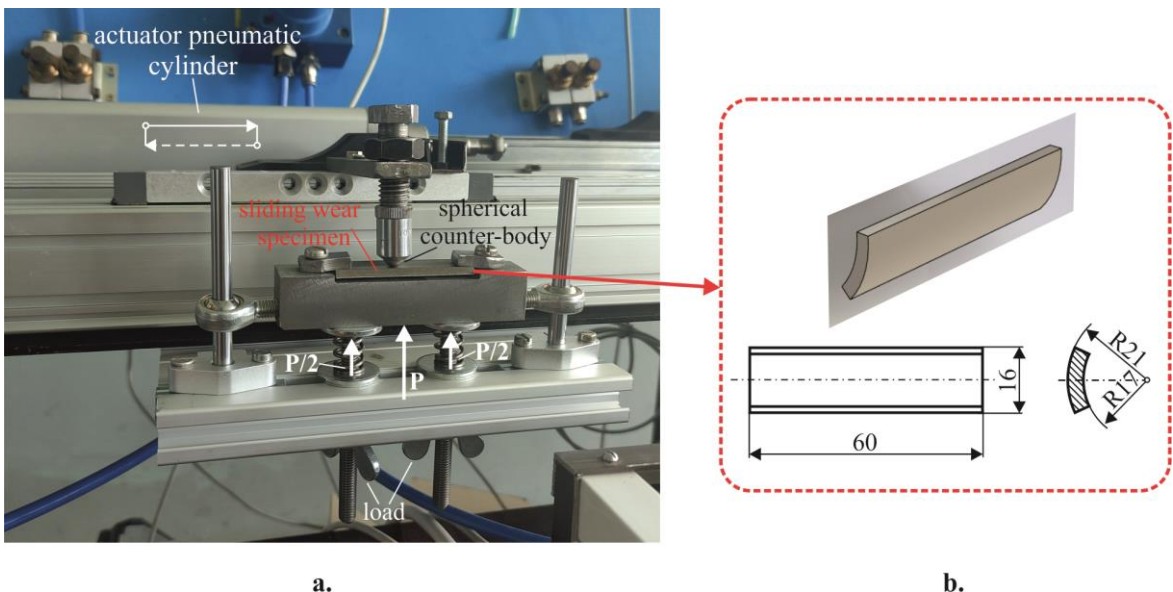

**Figure 3.** Sliding wear testing: (**a**) experimental setup and (**b**) specimen shape and sizes.

The experimental setup used to conduct the reciprocating sliding wear tests is shown in Figure 3. In this setup, the counter-body performs a reciprocating motion along the generatrix from the cylindrical surface in the specimen's plane of symmetry. This scheme replicates the loading present during cylinder line operation. In fact, this tribo-system is a modification of the well-known pin-on-disc scheme, in which rotary motion is replaced by a reciprocating motion. The pressing force P is set by means of two screw pairs symmetrically located with respect to the specimen and two screw springs. The movement of the counter-body is accomplished with a pneumatic cylinder.

## 3. Results and Discussion

### 3.1. Chemical Composition, Main Mechanical Characteristics, and Initial Microstructure

The chemical composition (in weight percentages) of the 321 SS is shown in Table 1. Figure 4 shows the results from the tensile tests: yield limit 300 MPa, tensile strength 596 MPa, and elongation 50%. The surface fractures of all specimens show typical tough-plastic destruction under the main tangential stresses acting in planes located at an angle of 45° from the tensile load direction. The hardness was established as 196 HB.

**Table 1.** Chemical composition (in wt%) of the tested 321 austenitic stainless steel.

| Fe | C | Si | Mn | P | S | Cr | Ni | Ti | Nb | Mo | Cu | Co | V | Others |
|---|---|---|---|---|---|---|---|---|---|---|---|---|---|---|
| 69.1 | 0.023 | 0.286 | 1.92 | 0.04 | 0.027 | 17.7 | 9.35 | 0.538 | 0.058 | 0.27 | 0.353 | 0.103 | 0.102 | balance |

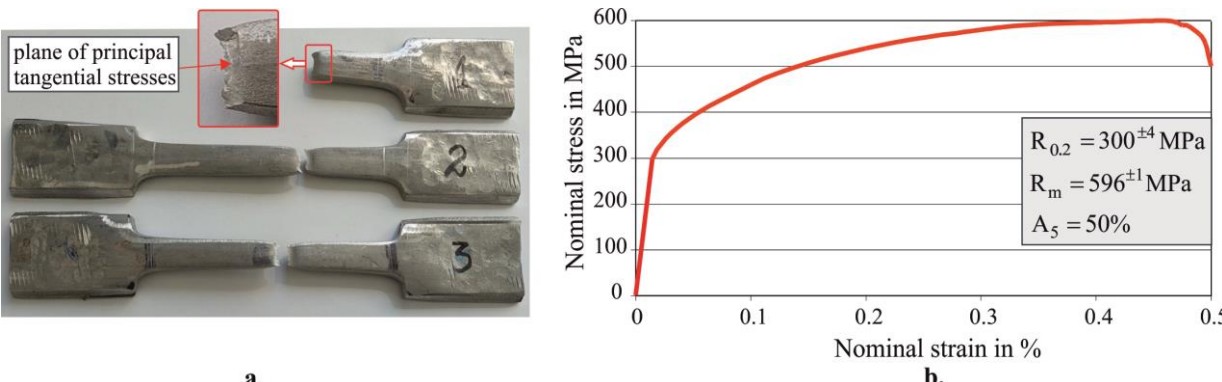

a.                                                                                        b.

**Figure 4.** Tensile test results: (**a**) fractured specimens and (**b**) engineering stress–strain diagram.

The phase analysis results for the as-received pipe are shown in Figure 5. The phase analysis was conducted in two areas of the pipe, namely the hole surface and front plane. The resulting diffractograms show no significant difference in the phase distribution across the two zones. An increased intensity for the austenite peaks (111) and (200) in the front plane was observed, which could have been due to the texturing of the steel immediately around the hole, a consequence of the pipe fabrication process.

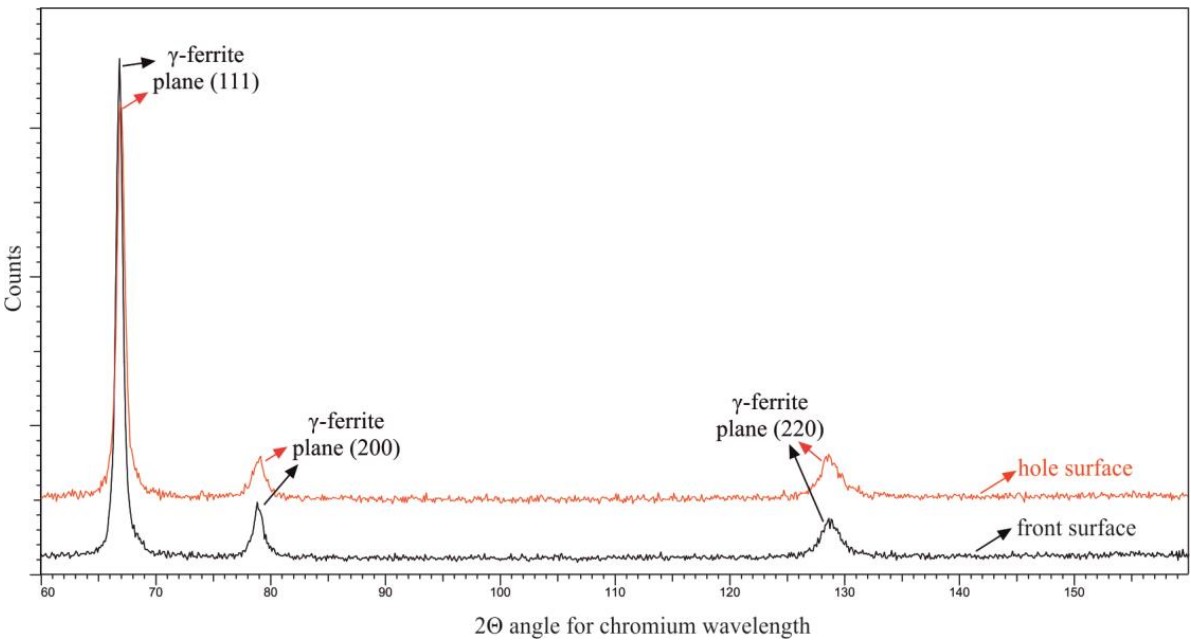

**Figure 5.** Phase analysis outcomes (as-received state).

The steel's microstructure is shown in Figure 6. Of interest are the surface around the hole and the subsurface layers around it at a depth of approximately 240 μm. At depths down to 60 μm, the structure is relatively homogeneous and includes columnar austenite grains with strongly pronounced carbide precipitates on their boundaries. The significant segregation of these carbides along grain boundaries in the surface layer is probably due to plastic deformation during the production process, which would have led to the dynamic aging of the material. At depths between 60 and 240 μm, the structure becomes rather inhomogeneous. Dispersed precipitates of titanium and niobium carbides are observed not only on the boundaries but also in the austenite grains themselves. Well-defined slip lines with significant precipitates along them blur the visible austenite grain boundaries.

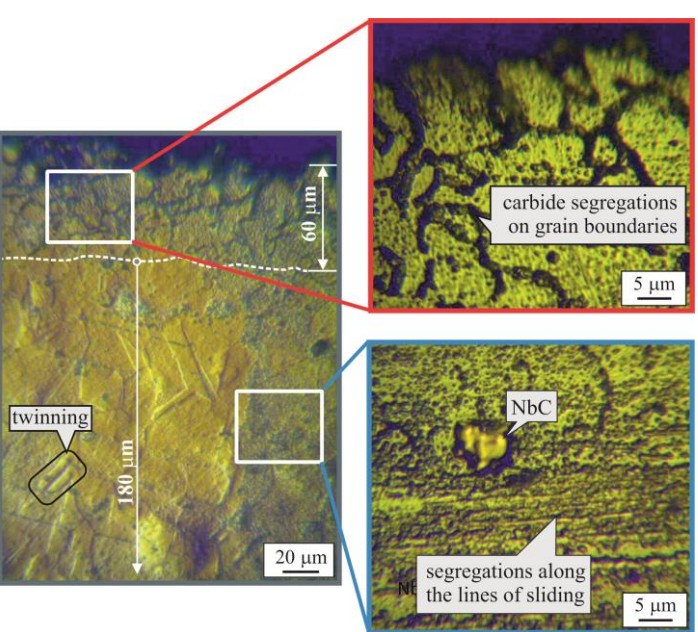

**Figure 6.** Microstructure in as-received state.

*3.2. Effect of DB on the Surface Roughness $R_a$ Height Parameter: One-Factor-at-a-Time Method*

3.2.1. Influence of the Burnishing Force and Diamond Insert Radius

The average value of the initial roughness parameter $R_a$ (after fine turning) was $R_a^{init} = 0.49$ μm and was achieved using a TCMX110204WP carbide cutting insert with the following turning parameters: feed rate $f = 0.05$ mm/tr, depth of cutting $a_p = 0.2$ mm, and cutting velocity $(40 - 60)$ m/min. Figure 7 shows the dependence of $R_a$ on the burnishing force and diamond insert radius. The minimum radius $(r = 2$ mm$)$ in combination with the minimum (below 50 N) and maximum (above 250 N) burnishing forces increases the initial roughness $R_a$, as does a combination of the mean radius $(r = 3$ mm$)$ and a burnishing force under 50 N. All other combinations decrease the roughness. This improvement is significant, except for combinations of the minimum radius and a burnishing force in the range of $(50 - 75)$ N.

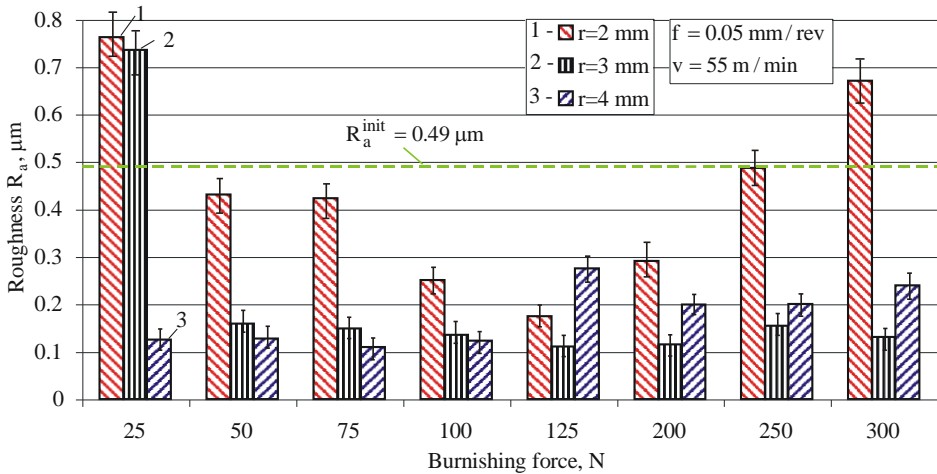

**Figure 7.** Influence of the burnishing force and radius on the $R_a$ roughness parameter.

3.2.2. Influence of the Feed Rate for Different Radius Values

Figure 8 shows the dependence of $R_a$ on the feed rate and diamond insert radius. All combinations improve the roughness. The improvement is significant, except for some com-

binations containing the minimum (f = 0.044 mm/rev) and maximum (f = 0.128 mm/rev) feed rates.

**Figure 8.** Influence of the feed rate at different radius magnitudes.

### 3.3. Effect of DB on the Surface Integrity: Planned Experiment and Optimization

Based on the one-factor-at-a-time experiment (see Figures 7 and 8), the following restrictions were used for the governing factors: the radius r was limited to 2, 3, and 4 mm, the burnishing force $F_b$ was limited to the interval $80 \leq F_b \leq 240$ N, and the feed rate f had to be a value from the interval $0.05 \leq f \leq 0.1$ mm/rev. Table 2 contains the governing factor levels. The transformation from natural $\widetilde{x}_i$ into coded $x_i$ coordinates was carried out according to the formula:

$$x_i = \frac{2(\widetilde{x}_i - \widetilde{x}_{0,i})}{\widetilde{x}_{max,i} - \widetilde{x}_{min,i}},\tag{1}$$

where $\widetilde{x}_{max,i}$, $\widetilde{x}_{0,i}$, and $\widetilde{x}_{min,i}$ are the upper, middle, and lower levels of the i-th factor in natural coordinates, respectively.

**Table 2.** Governing factor levels.

| Governing Factors | | | | Levels | | | | |
| Natural | Codded | | Natural | | | | Coded | |
|---|---|---|---|---|---|---|---|---|
| Diamond radius, mm | r | $x_1$ | 2 | 3 | 4 | | | |
| Burnishing force, N | $F_b$ | $x_2$ | 80 | 160 | 240 | −1 | 0 | 1 |
| Feed rate, mm/rev | f | $x_3$ | 0.05 | 0.075 | 0.1 | | | |

Based on the justification made in Section 1, the objective functions are: $Y_{R_a}$, $Y_{R_q}$, $Y_{R_v}$, $Y_{R_{sk}}$, $Y_{R_{ku}}$, $Y_{HV}$, $Y_{\sigma_a^{res}}$, and $Y_{\sigma_t^{res}}$. As the governing factors change on three levels, an optimal compositional plan of the second order was chosen for the planned experiment (Table 3).

Because the strain-induced $\alpha'$-martensite content in the surface layer was below 5%, the measurement of the surface residual stresses was carried out for the γ-Fe phase. The measured diffraction profiles, as determined by the Pearson VII method, of the γ-Fe {220} plane exhibited a maximum at $2\theta \approx 128.8^{\circ}$ for the filtered VKα radiation used. The Winholtz and Cohen method with X-ray elastic constants $s_1 = -1.352$ TPa$^{-1}$ and $\frac{1}{2}s_2 = 6.182$ TPa$^{-1}$ was applied (γ-phase). A 2θ range of 124°–133° for γ-Fe was used, with 2θ steps of 0.5° and a tilt defined by $\sin^2 \psi = 0, 0.1, 0.2, 0.3, 0.4$, and 0.5 for both positive and negative $\psi$ angle values.

**Table 3.** Experimental design and results.

| Exp. Point | $x_1$ | $x_2$ | $x_3$ | $R_a$, μm | $Y_{R_a}$, μm | $R_q$, μm | $Y_{R_q}$, μm | $R_v$, μm | $Y_{R_v}$, μm | $R_{sk}$ | $Y_{R_{sk}}$ |
|---|---|---|---|---|---|---|---|---|---|---|---|
| 1 | −1 | −1 | −1 | 0.266 | 0.2621 | 0.319 | 0.3134 | 0.595 | 0.5579 | 0.104 | 0.1060 |
| 2 | 1 | −1 | −1 | 0.175 | 0.1853 | 0.245 | 0.2560 | 0.825 | 0.8849 | −0.468 | −0.5498 |
| 3 | −1 | 1 | −1 | 0.252 | 0.2432 | 0.299 | 0.2943 | 0.667 | 0.7122 | 0.070 | 0.0824 |
| 4 | 1 | 1 | −1 | 0.071 | 0.055 | 0.094 | 0.0703 | 0.400 | 0.3852 | −0.489 | −0.5734 |
| 5 | −1 | −1 | 1 | 0.189 | 0.1958 | 0.232 | 0.2401 | 0.742 | 0.8204 | −0.322 | −0.3822 |
| 6 | 1 | −1 | 1 | 0.252 | 0.2516 | 0.340 | 0.3292 | 1.249 | 1.1474 | −1.136 | −1.0380 |
| 7 | −1 | 1 | 1 | 0.178 | 0.1769 | 0.219 | 0.2210 | 0.446 | 0.4497 | 0.530 | 0.5706 |
| 8 | 1 | 1 | 1 | 0.108 | 0.1212 | 0.125 | 0.1436 | 0.142 | 0.1227 | −0.090 | −0.0852 |
| 9 | −1 | 0 | 0 | 0.119 | 0.1376 | 0.157 | 0.1644 | 0.296 | 0.2863 | 0.089 | 0.0942 |
| 10 | 1 | 0 | 0 | 0.067 | 0.0714 | 0.085 | 0.0970 | 0.291 | 0.2863 | −0.625 | −0.5616 |
| 11 | 0 | −1 | 0 | 0.166 | 0.1418 | 0.197 | 0.1819 | 0.801 | 0.8158 | −0.872 | −0.9135 |
| 12 | 0 | 1 | 0 | 0.066 | 0.0672 | 0.084 | 0.0795 | 0.381 | 0.3806 | −0.392 | −0.4489 |
| 13 | 0 | 0 | −1 | 0.169 | 0.1864 | 0.220 | 0.2335 | 0.889 | 0.9468 | −0.797 | −0.6812 |
| 14 | 0 | 0 | 1 | 0.204 | 0.1864 | 0.242 | 0.2335 | 1.019 | 0.9468 | −0.664 | −0.6812 |

| Exp. point | $R_{ku}$ | $Y_{R_{ku}}$ | $H_v$ | $H_v$ scattering | $Y_{HV}$ | $\sigma_a^{res}$, MPa | $\sigma_a^{res}$ meas. error, MPa | $Y_{\sigma_a^{res}}$, MPa | $\sigma_t^{res}$, MPa | $\sigma_t^{res}$ meas. error, MPa | $Y_{\sigma_t^{res}}$, MPa |
|---|---|---|---|---|---|---|---|---|---|---|---|
| 1 | 2.817 | 2.8816 | 467 | ±24.5 | 470.67 | −428.5 | 50.3 | −454.7 | −79.1 | 25.5 | −53.6 |
| 2 | 4.571 | 4.7756 | 457 | ±31 | 464.32 | −244.0 | 59.7 | −251.6 | 19.2 | 20.8 | −9.4 |
| 3 | 2.213 | 2.0266 | 490 | ±43 | 494.92 | −431.5 | 57.1 | −436.8 | 117.3 | 76.9 | 119.4 |
| 4 | 3.651 | 3.9206 | 441 | ±21 | 440.07 | −325.8 | 62.8 | −305.7 | 82.9 | 39.4 | 101.4 |
| 5 | 2.737 | 2.8816 | 451 | ±38 | 451.47 | −415.9 | 37.4 | −435.9 | −29.0 | 44.5 | −46.5 |
| 6 | 5.271 | 4.7756 | 452 | ±24.5 | 445.12 | −155.7 | 50.4 | −150.3 | 101.7 | 48 | 100.4 |
| 7 | 2.165 | 2.0266 | 481 | ±26.5 | 475.72 | −489.5 | 83.8 | −481.9 | −20.1 | 60.8 | 9.5 |
| 8 | 3.693 | 3.9206 | 424 | ±37 | 420.87 | −294.4 | 61 | −268.2 | 118.8 | 59 | 94.3 |
| 9 | 2.950 | 3.16.72 | 477 | ±26 | 473.20 | −577.0 | 38.3 | −532.9 | 28.5 | 61.7 | −11.2 |
| 10 | 5.166 | 5.0612 | 439 | ±36 | 442.60 | −280.5 | 30.7 | −324.5 | 17.4 | 46.7 | 53.2 |
| 11 | 3.747 | 3.8286 | 410 | ±36.5 | 410.25 | −198.0 | 78.7 | −149.4 | −0.6 | 32.4 | 21.3 |
| 12 | 3.146 | 2.9736 | 400 | ±39.5 | 410.25 | −151.0 | 50.2 | −199.5 | 130.7 | 39.7 | 104.8 |
| 13 | 4.340 | 4.1142 | 440 | ±27 | 419.85 | −197.0 | 28 | −177.9 | 63.6 | 21.8 | 44.6 |
| 14 | 4.001 | 4.1142 | 391 | ±43 | 400.65 | −130.7 | 67.7 | −149.7 | 21.9 | 35.6 | 44.6 |

The experimental results for the chosen roughness parameters, surface micro-hardness, surface residual hoop, and axial stresses are shown in Table 3.

Regression analyses were also carried out. The significance of the regression coefficients was determined at the p = 0.05 level. Given the chosen experimental design (three levels for each factor), the approximating polynomials were of second order:

$$Y_k(\{X\}) = b_0 + \sum_{i=1}^{m} b_i x_i + \sum_{i=1}^{m-1} \sum_{j=i-1}^{m} b_{ij} x_i x_j + \sum_{i=1}^{m} b_{ii} x_i^2, \ k = 1, 2, \ldots q, \tag{2}$$

where $\{X\}$ is the vector of the governing factors, m is the number of governing factors, and q is the number of objective functions.

QStatLab software was used to conduct the analyses [34]. The regression coefficients are shown in Table 4. The values of the objective functions calculated using Equation (2) for the experimental points of the plan are shown in Table 3. The comparison between the experimental results and those predicted by the models (at the experimental points) indicates good agreement.

**Table 4.** Coefficients of regression.

| $Y_k$ | $b_0$ | $b_1$ | $b_2$ | $b_3$ | $b_{11}$ | $b_{22}$ | $b_{33}$ | $b_{12}$ | $b_{13}$ | $b_{23}$ |
|---|---|---|---|---|---|---|---|---|---|---|
| $Y_{R_a}$ | 0.1045 | −0.0331 | −0.0373 | 0 | 0 | 0 | 0.0819 | −0.0279 | 0.03312 | 0 |
| $Y_{R_q}$ | 0.13075 | −0.0337 | −0.0512 | 0 | 0 | 0 | 0.10275 | −0.0416 | 0.03662 | 0 |
| $Y_{R_v}$ | 0.59815 | 0 | −0.2176 | 0 | −0.3118 | 0 | 0.34869 | −0.1635 | 0 | −0.1312 |
| $Y_{R_{sk}}$ | −0.6812 | −0.3279 | 0.2323 | 0 | 0.4475 | 0 | 0 | 0 | 0 | 0.2441 |
| $Y_{R_{ku}}$ | 4.11425 | 0.947 | −0.4275 | 0 | 0 | −0.7131 | 0 | 0 | 0 | 0 |
| $Y_{HV}$ | 410.25 | −15.3 | 0 | −9.6 | 47.65 | 0 | 0 | −12.125 | 0 | 0 |
| $Y_{\sigma_a^{res}}$ | −209.47 | 104.2 | −25.01 | 14.06 | −219.28 | 34.9687 | 45.6187 | −17.987 | 20.637 | −15.937 |
| $Y_{\sigma_t^{res}}$ | 44.6846 | 32.24 | 41.74 | 0 | −23.669 | 18.4308 | 0 | −15.562 | 25.7125 | −29.262 |

The optimization task was set as follows. The vector of the objective functions was:

$$\left\{ \vec{Y}(\{X\}) \right\} = \left[ Y_{R_a} \ Y_{R_q} \ Y_{R_v} \ Y_{R_{sk}} \ Y_{R_{ku}} \ Y_{HV} \ Y_{\sigma_a^{res}} \ Y_{\sigma_t^{res}} \right]^T, \tag{3}$$

where:

$$\begin{cases} Y_{R_a} \to \min, \ Y_{R_q} \to \min, \ Y_{R_v} \to \max, \ Y_{R_{sk}} \to \min, \\ Y_{R_{ku}} \to \max, \ Y_{HV} \to \max, \ Y_{\sigma_a^{res}} \to \min, \ Y_{\sigma_t^{res}} \to \max \end{cases}, \tag{4}$$

$$\{X\} = [x_1 \ x_2 \ x_3]^T \in \Gamma_x, \tag{5}$$

and $\Gamma_x$ is the space of the governing factors $x_i$.

The governing factor limitations are shown in Table 2. Constraints are imposed on two of the objective functions, namely skewness $Y_{S_{sk}}$ and kurtosis $Y_{S_{ku}}$, for which limitations arise from the functional purpose of the treated surface; that is, the desire for maximum wear resistance in the presence of a lubricant [22,24]:

$$Y_{S_{sk}} < 0 \ \text{and} \ Y_{S_{ku}} > 3. \tag{6}$$

The vector $\{X^*\}$ must be found so that the objective function magnitudes $Y_k(\{X^*\})$ satisfy (4) and (6), and

$$\{X^*\} = [x_1^* \ x_2^* \ x_3^*]^T \in \Gamma_x,$$

where $x_1^*$, $x_2^*$, and $x_3^*$ are the governing factor-compromised optimal values.

The defined multi-objective optimization task was solved via the Pareto-optimal solutions approach. Decisions were made with a non-dominated sorting genetic algorithm (NSGA-II) [35] available in QstatLab. Because the nominal radius sizes of the diamond insert were integers (2, 3, and 4 mm), additional restrictions on the governing factor $x_1$ were imposed. Taking into account both Equation (1) and the fact that the deviation from the nominal size of the radius is usually ±0.1 mm, these restrictions were: for r =2 mm, $x_1$ was confined to the interval $-1 \le x_1 \le -0.9$; for r =3 mm, $x_1$ was limited to the interval $-0.1 \le x_1 \le 0.1$; for r =4 mm, $x_1$ was restricted to $0.9 \le x_1 \le 1$. In this manner, three Pareto-optimal solution sets (for each of the three radius sizes) were obtained, each containing 50 solutions proposed by QStatLab. From these sets, one solution was selected for each radius (Table 5). Specimen holes (three specimens for each of the three radii) with a length of 60 mm were then diamond-burnished using the compromise optimal values of the governing factors. The measured values of the selected parameters, namely roughness, microhardness, and residual stresses, are shown in Table 6, where each value was obtained as the arithmetic mean of the values for the three specimens at a specified radius. The comparison with Table 5 shows a good agreement of the measured values with those of the optimization.

**Table 5.** Selected solutions from the Pareto fronts.

| № | r, mm | $F_b$, N | f, mm/tr | $R_a$, μm | $R_q$, μm | $R_v$, μm | $R_{sk}$ | $R_{ku}$ | HV | $\sigma_t^{res}$, MPa | $\sigma_a^{res}$, MPa |
|---|---|---|---|---|---|---|---|---|---|---|---|
| 1 | 2 | 80 | 0.088 | 0.163 | 0.197 | 0.601 | −0.314 | 2.893 | 453.8 | 46.8 | −458.9 |
| 2 | 3 | 221 | 0.078 | 0.077 | 0.093 | 0.426 | −0.485 | 3.372 | 409.3 | −84.9 | −207.7 |
| 3 | 4 | 218 | 0.054 | 0.057 | 0.0752 | 0.355 | −0.549 | 4.379 | 408.3 | 77.9 | −321.5 |

**Table 6.** Surface hole characteristics of specimens treated via optimized DB and grinding.

| Group № | Finishing | $R_a$, μm | $R_q$, μm | $R_v$, μm | $R_{sk}$ | $R_{ku}$ | HV | $\sigma_t^{res}$, MPa | $\sigma_a^{res}$, MPa |
|---|---|---|---|---|---|---|---|---|---|
| 1 | DB with r = 2 mm | 0.228 | 0.304 | 0.891 | −0.193 | 3.574 | 470.1 | 26.2 | −588.1 |
| 2 | DB with r = 3 mm | 0.147 | 0.197 | 0.769 | −0.870 | 5.088 | 425.7 | −80.3 | −178.3 |
| 3 | DB with r = 4 mm | 0.102 | 0.125 | 0.286 | −0.010 | 2.640 | 420.9 | 88.3 | −420.3 |
| 4 | grinding | 0.387 | 0.351 | 2.262 | −1.442 | 5.633 | 417.9 | 46.1 | −329.7 |

*3.4. Reciprocating Sliding Wear Resistance of the Treated Surfaces*

3.4.1. Specimen Treatments and Designations

To prove the effectiveness of DB as a finishing method, a comparison of DB with the traditionally used grinding method was made. The holes of three samples were ground on a STUDER S33 grinding machine, utilizing a tool made of cubic boron nitride. The tool diameter was 30 mm. The rotational frequencies of the workpiece and the tool were 200 min$^{-1}$ and 35,000 min$^{-1}$, respectively, whereas the depth of cut and the feed rate were 0.01 mm and 1000 mm/min, respectively.

The measured values of the selected parameters for the roughness, micro-hardness, and residual stresses are shown in Table 6, where each value was obtained as an arithmetic mean of the values obtained for three samples under a specific finishing type, namely grinding or optimized DB, for r = 2 mm, r = 3 mm, and r = 4 mm. The amplitude parameters for the roughness ($R_a$, $R_q$, and $R_v$) for all groups of diamond-burnished specimens are lower than those of the ground specimen. These parameters decrease as the diamond insert radius increases. All finishing processes result in negative skewness with a relatively high kurtosis parameter. Obviously, the grinding leads to the lowest surface micro-hardness, a consequence of the small amount of coldwork inherent in cutting finishing processes. DB, implemented with radius r = 2 mm, provides the largest surface compressive axial residual stresses. Despite the fact that the grinding introduces relatively large axial compressive residual stresses in the surface layer, the depth of the compression zone is significantly smaller than that for DB (Figure 9).

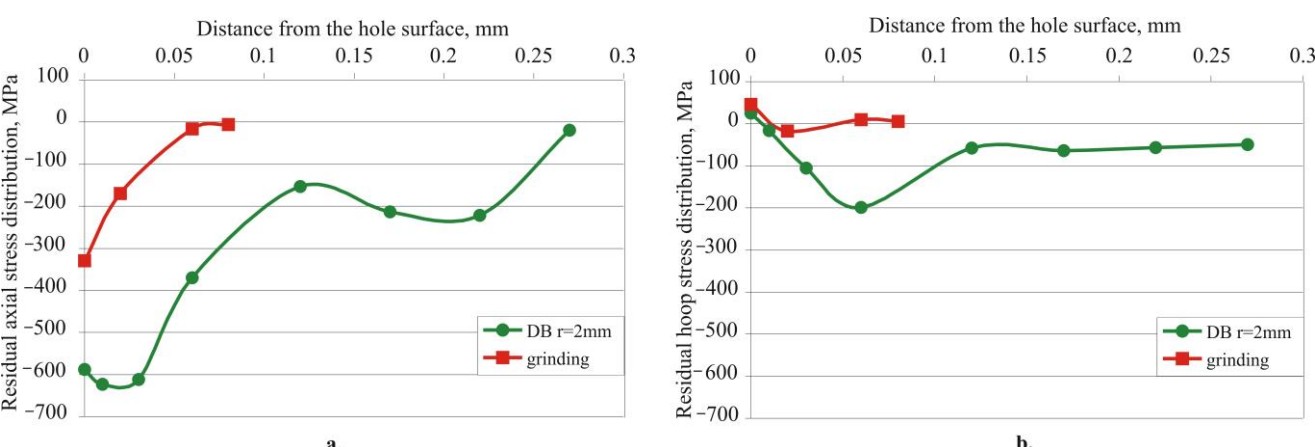

**Figure 9.** Residual stress distribution introduced by optimized DB (r = 2 mm) and grinding: (**a**) axial and (**b**) hoop.



Samples were cut from the machined cylinders to reciprocate sliding wear tests (see Figure 1, step 7). The samples were classified into four groups: DB1, DB2, DB3, and G4. The samples in DB1, DB2, and DB3 were diamond-burnished using diamond insert radii 2, 3, and 4 mm, respectively. The samples in G4 were ground. Each group contained six specimens: three for the dry friction mode and three for the boundary lubrication regime. For each group, the final result for the mass wear $\Delta m_i$ for a given friction path $L_i$ was calculated as the arithmetic mean of the mass wear values for the three relevant samples.

3.4.2. Reciprocating Sliding Wear Tests

The pressing force P for the sliding wear tests was chosen based on the following condition: the maximum contact equivalent stress at the point on the surface of the specimen in contact with the spherical surface of the counter-body should be equal to the yield strength (see Figure 4) of the base material. The finite element method (FEM) in Abaqus Standard software v. 6.12-1 and implicit analysis were used to pinpoint P. The FE model is shown in Figure 10. Using the experimental setup (see Figure 3), a quarter of the counter body–specimen–fixture–support system was modeled (due to the double symmetry, the counter body was positioned in the middle of the specimen). Elastic behavior of hardened carbon steel was assumed for the fixture and support. The spherical counter body was modeled as an analytical rigid body, and the specimen was modeled as an elastic-plastic body with a material constitutive model (according to Figure 4) and isotropic strain hardening because the yield surface was not going to move in the stress space. The roughness and residual stresses in the surface layer of the specimen were neglected. A total of three master–slave contacts were defined. Displacement of the rigid body reference point (RP) along the *x*-axis was assigned, varying linearly in pseudo-time. P was then measured as the reaction (in x direction) of the RP. First, linear finite elements of the type C3D8R were selected. It was then determined that the sizes of the finite elements contacting the rigid body along the three directions were approximately 0.08 mm. Finally, the maximum equivalent stress of 300 MPa at the contact point between the rigid body and the specimen was obtained for a pressing force of approximately 12 N. Therefore, the reciprocating sliding wear tests were conducted with a P of 12 N.

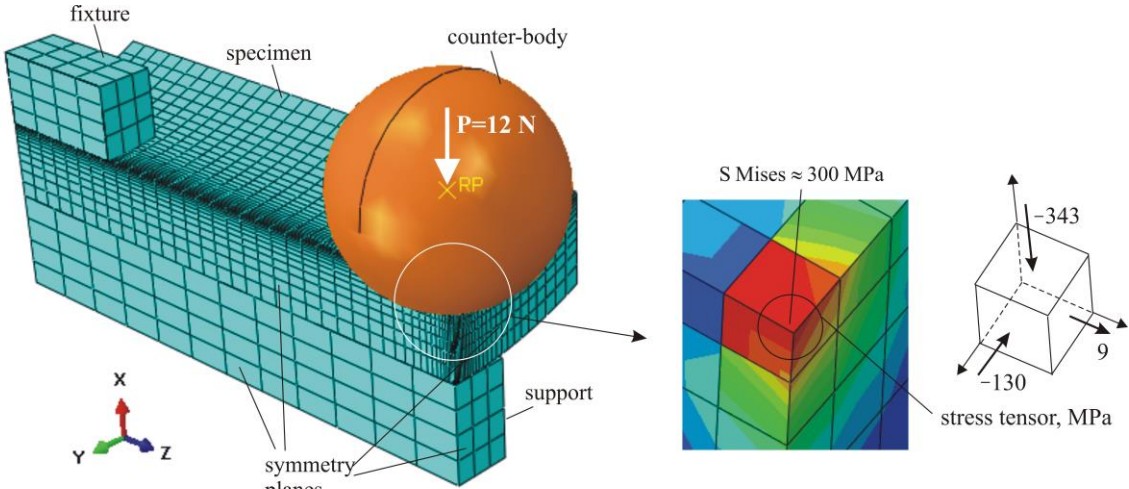

**Figure 10.** Finite element determination of the pressing force magnitude.

The kinetic curves of mass wear values for all groups of specimens in the dry friction regime are depicted in Figure 11. For all values of the friction path, the mass wear is least for the diamond-burnished specimens in group DB1, followed by the specimens in group G4, with ground holes. The mass wear found for the other two groups was greater despite their roughness amplitude parameters being lower than those for group DB1 (see Table 6). At the same time, the most favorable combination of the shape parameters $R_{sk}$ and $R_{ku}$ in

terms of friction was measured for specimens from group DB2. However, these parameters were of decisive importance to retain the lubricant in the boundary and mixed friction modes, respectively [11,22,24]. In the dry friction mode, the abrasive wear mechanism generally dominates, which, to the greatest extent, depends on the physical-mechanical characteristics of SI: surface micro-hardness HV, plus the surface axial ($\sigma_a^{res}$) and hoop ($\sigma_t^{res}$) residual stresses [11]. The highest micro-hardness (470 HV) and maximum surface compressive axial residual stresses were measured ($\sigma_a^{res} = -588.1$ MPa) for the DB1 group samples (Table 6). The axial residual stress profile (Figure 9a) confirms the presence of a compressive zone with a depth greater than 0.25 mm and maximum stress at a depth of approximately 0.05 mm. The surface residual hoop stress is relatively small tensile (26.2 MPa), but at a depth of approximately 0.05 mm, it reaches $-200$ MPa (Figure 9b). These results confirm the importance of a greater amount of cold work on the surface layer in reducing the wear in the dry friction mode [11].

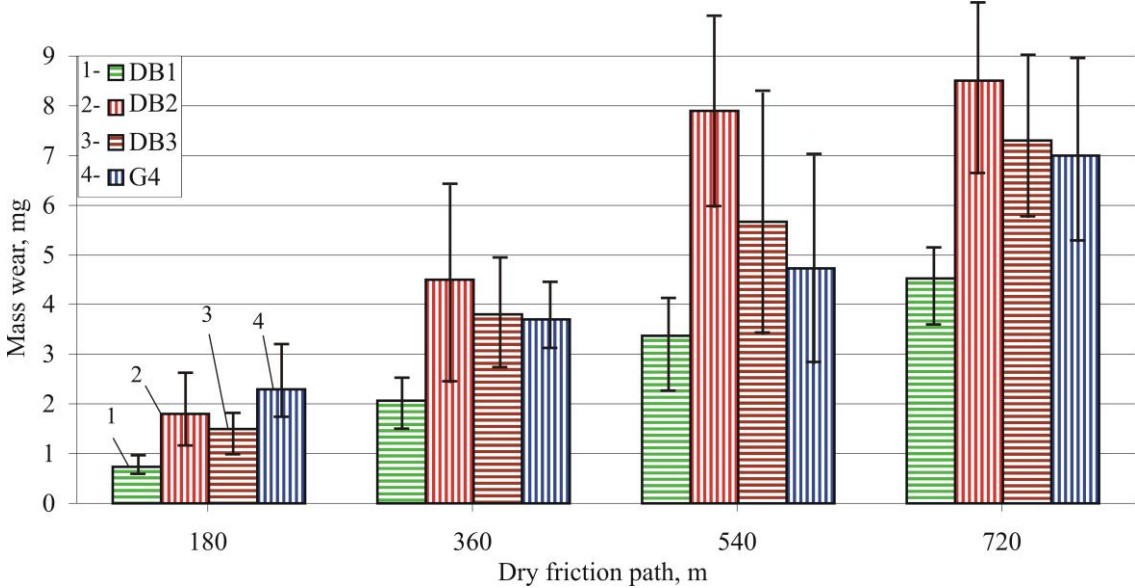

**Figure 11.** Alteration of mass wear depending on the friction path at dry friction mode.

In order to predict wear over time, it is necessary to determine the wear rate trends for the corresponding sliding wear specimens. The average wear rate and the corresponding trend lines are shown in Figure 12. For the researched friction path, the samples from group DB1 show the lowest wear rate. However, the trend lines for all groups of diamond-burnished specimens (DB1, DB2, and DB3) and the trend line for the ground specimens (G4) are opposite in character. The wear rates of the diamond-burnished specimens increase smoothly up the friction path for approximately 540 m, after which they also smoothly decrease. For the ground samples in G4, the opposite is true: up to 180 m along the friction path, the wear rate is maximum, then it smoothly decreases until approximately 540 m along the friction path, at which point it starts to increase. It can be assumed that for diamond-burnished specimens, the running-in stage ends along the friction path of approximately 540 m, after which the specimens reach an equilibrium surface texture. The increasing wear rate after 540 m of friction path for the ground samples (G4) is an indication that their running-in stage has not ended. The observed phenomenon confirms Korzynski et al.'s [24] finding that DB provides a surface texture close to the equilibrium one obtained at the end of the running-in stage on surfaces treated by cutting only.

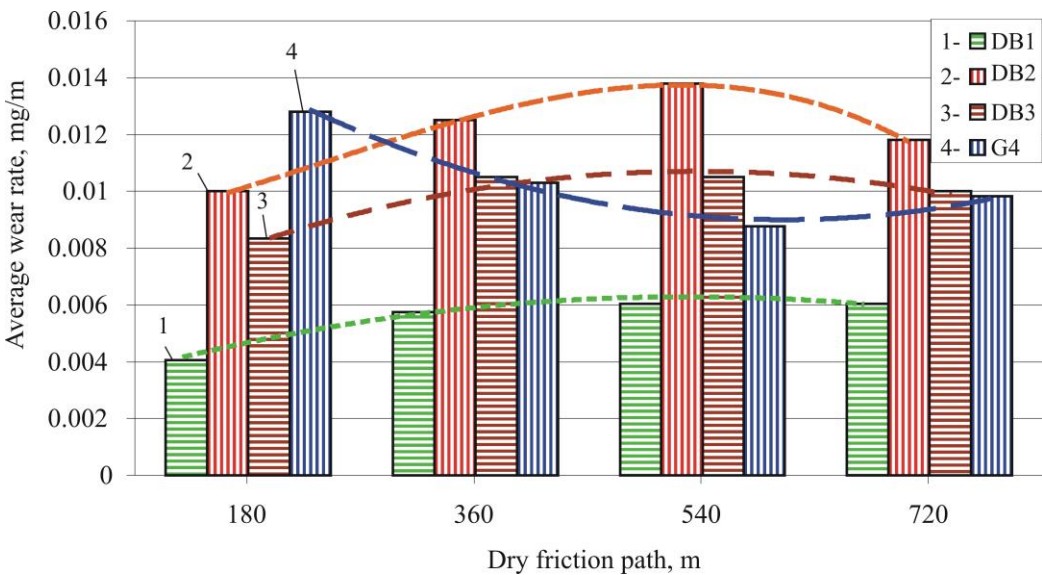

**Figure 12.** Average wear rate depending on the friction path at dry friction mode.

The kinetic curves of mass wear for all groups of specimens in the boundary lubrication friction regime are depicted in Figure 13. Under this friction mode, mass wear is the least for the diamond-burnished specimens in group DB1. At the end of the friction path (1000 m), the mass wear of the ground samples in G4 is greatest. For the other two groups of diamond-burnished specimens (DB1 and DB3), the wear develops at a changing rate depending on the friction path. The average wear rate and the corresponding trend lines of the sample groups are shown in Figure 14. The trend lines for the wear rates of the samples from group DB1 increase smoothly until reaching 750 m along the friction path, after which they maintain almost constant values. The wear rate trend lines for the diamond-burnished specimens from groups DB2 and DB3 have a variable character (Figure 14). For group DB2, the wear rate markedly decreases in the interval 750–1000 m. As a result, the DB2 group specimens come in second by the friction path end (Figure 13). The relatively high wear resistance in the boundary friction mode of the specimens in this group can be explained by their more favorable combination of shape parameters ($R_{sk} = -0.87$, $R_{ku} = 5.088$) compared to the other two groups of diamond-burnished specimens (see Table 6). This combination describes a micro-roughness profile dominated by deep valleys and sharp peaks, which improves lubricant retention and reduces friction. As found in other studies on sliding wear [11], in the boundary friction regime, the functional importance of the SI geometric characteristics exceeds that of these characteristics under the dry friction regime. On the other hand, the smaller amount of wear at the end of the investigated friction paths of all groups of diamond-burnished specimens confirms the beneficial effect of cold work and the importance of lower values for the roughness amplitude parameters. Despite the fact that grinding provides the most favorable combination of shape parameters ($R_{sk} = -1.442$, $R_{ku} = 5.633$; see Table 6), the wear rate of the ground samples increases throughout the friction path. Although it remains relatively weak until reaching 500 m along the friction path, the wear rate increases significantly after this point.

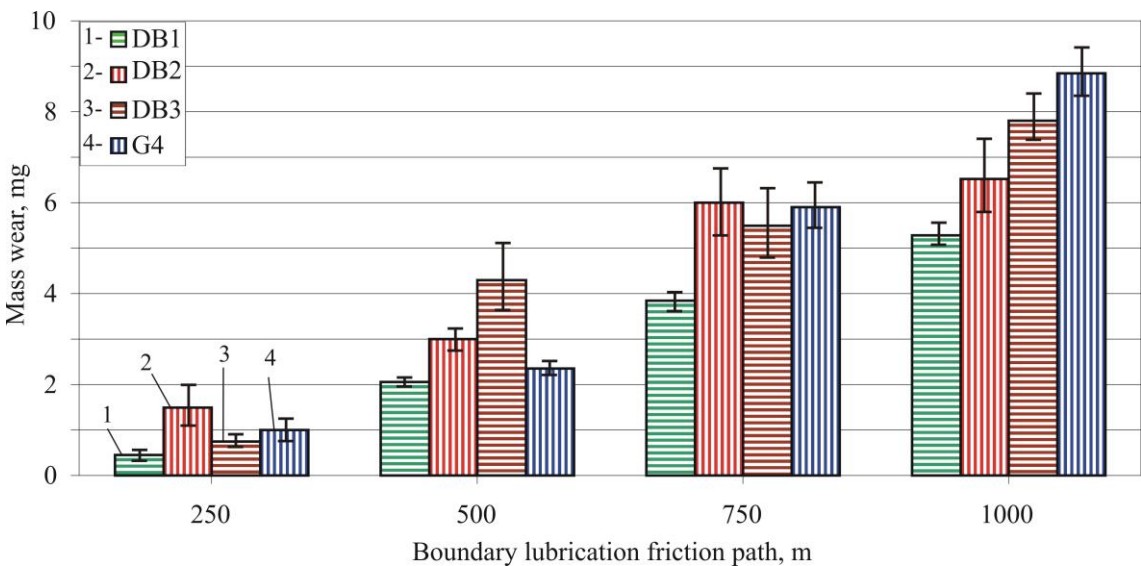

**Figure 13.** Alteration of mass wear depending on the friction path at boundary lubrication friction mode.

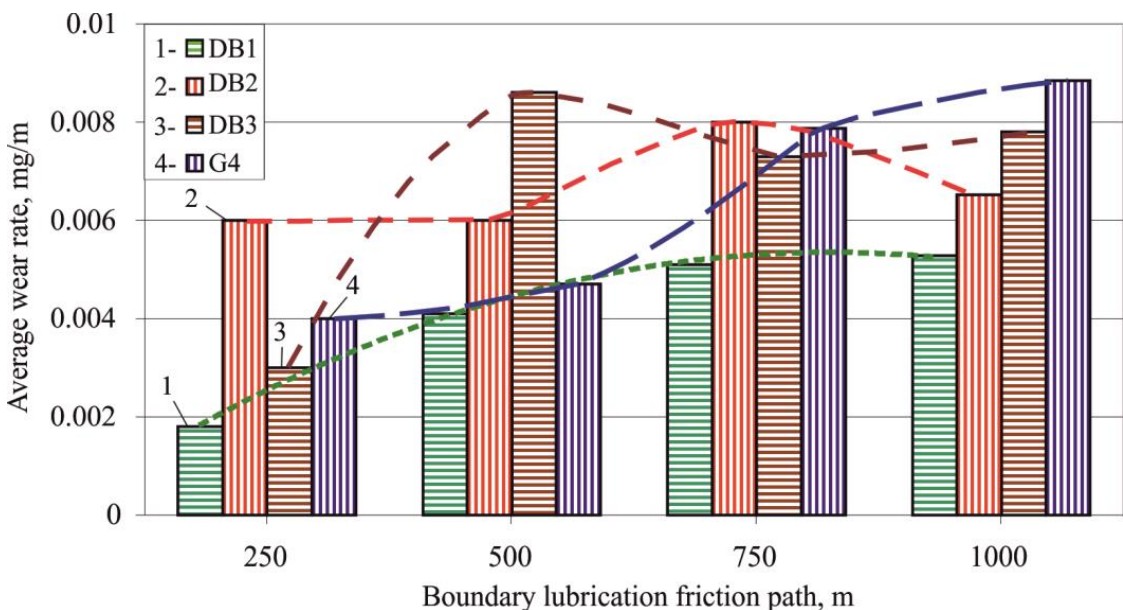

**Figure 14.** Average wear rate depending on the friction path at boundary lubrication friction mode.

The results obtained for the reciprocating sliding wear tests for both the dry and boundary lubrication friction regimes show that from the point of view of minimizing the wear on cylinder lines made of AISI 321 SS, the most suitable finishing process is that used for group DB1.

The morphology of the worn surfaces permits the evaluation of the wear mechanism. Figure 15 shows the worn surfaces created under the dry friction mode on the DB1 and G4 specimens. Abrasive scratches, oxidized zones, and adhesive pits are observed on these surfaces. This set of morphological characteristics indicates a mixed-wear mechanism for both surfaces featuring abrasive and adhesive wear. In general, larger oxidized areas and more pronounced adhesive pits are observed on the ground surface of the specimen from group G4 (Figure 15b). The lower surface hardness and higher roughness of these ground surfaces result in greater frictional forces; hence, a greater amount of generated

heat favoring oxidation and adhesion. Thus, the adhesion mechanism of wear is more pronounced for group G4.

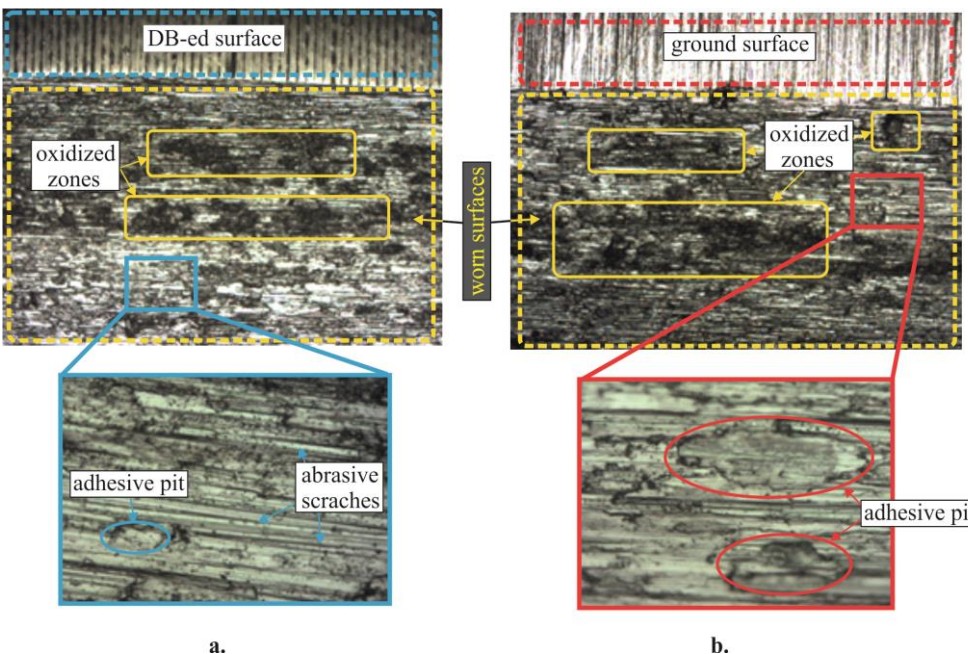

**Figure 15.** Worn surface under dry friction regime: (**a**) specimen DB1 and (**b**) specimen G4.

Figure 16 shows the DB1 and G4 specimen surfaces after sliding wear under the boundary lubrication friction mode. The morphology of the worn surfaces is mainly characterized by abrasive scratches, which are deeper for the ground surface (G4). The observed pits for both types of surfaces have a different genesis than those obtained under the dry friction condition. The reciprocating movement of the spherical counter-body in the presence of the lubricant causes contact fatigue on the surface layers of the samples. This phenomenon is manifested by the appearance of cracks into which lubricant falls: as a result of the increased pressure in the cracks, particles are released from the surface layer. The contact fatigue is more pronounced in the ground sample (G4). In general, in both types of specimens, abrasive wear predominates.

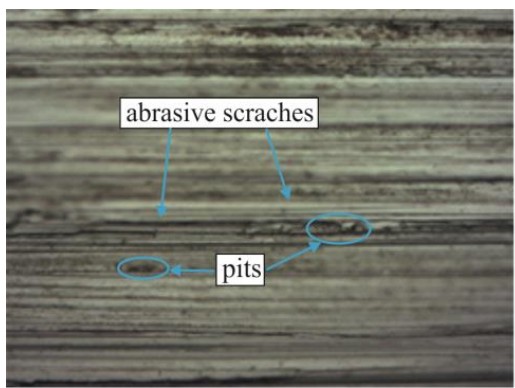
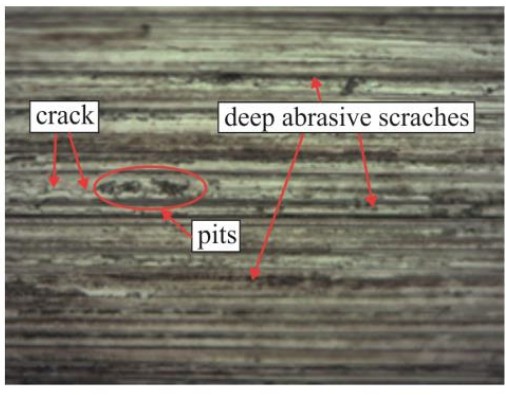

a.                                    b.

**Figure 16.** Worn surface under boundary lubrication friction mode: (**a**) specimen DB1 and (**b**) specimen G4.

## 4. Conclusions

When processing the holes of cylinder lines made of austenitic SS, material removal finishing processes are usually used: honing (for long holes) or grinding (for short-length holes). These processes require specialized machines and equipment. A new cold-working technology for finishing the holes of short-length cylinder lines exhibiting improved wear resistance was developed in this study. The proposed technology is based on the multi-objective optimization of the DB process. DB was implemented via a simple burnishing device for the same machine tool on which preliminary treatments were carried out. Thus, this technology increases productivity and reduces the cost of the product. The major new findings concerning the nature of the proposed technology of this study are:

- The DB of holes in AISI 321 SS, performed with a diamond insert radius $r = 2$ mm, burnishing force $F_b = 80$ N, and feed rate $f = 0.088$ mm/rev, significantly improves wear resistance. This optimal DB process maximizes the reciprocating sliding wear resistance of the hole surface in both dry friction and boundary lubrication friction modes.
- The wear rate trend line for sample holes processed with the optimal DB process decreased at the end of the friction path under both friction modes. The wear rate trend line for ground samples showed an opposite trend for both modes of friction. These experimental results show that to minimize the wear of holes in AISI 321 SS, a more pronounced cold work effect and minimum values of the roughness amplitude parameters are required.
- The developed DB device with elastic beam is applicable for holes with a wide range of variation in their diameters.

**Author Contributions:** Conceptualization, G.D. and J.M.; methodology, G.D. and J.M.; software, G.D. and J.M.; validation, G.D., J.M., A.A. and V.D.; formal analysis, G.D. and J.M.; investigation, A.A., V.D., Y.A., G.D., S.V. and J.M.; resources, G.D. and J.M.; data curation, G.D. and J.M.; writing—original draft preparation, G.D. and J.M.; writing—review and editing, G.D. and J.M.; visualization, G.D., J.M., A.A. and V.D.; supervision, G.D. and J.M.; project administration, G.D. and J.M.; funding acquisition, G.D. and J.M. All authors have read and agreed to the published version of the manuscript.

**Funding:** This research was supported by the European Regional Development Fund within the OP "Science and Education for Smart Growth 2014–2020", Project CoC "Smart Mechatronics, Eco- and Energy Saving Systems and Technologies", No.BG05M2OP001-1.002-0023.

**Data Availability Statement:** Not applicable.

**Conflicts of Interest:** The authors declare no conflict of interest.

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
