# Peer review of "Modeling and Optimization of Surface Integrity and Sliding Wear Resistance of Diamond-Burnished Holes in Austenitic Stainless Steel Cylinder Lines"

_machines, doi:10.3390/machines11090872_

Round 1
Reviewer 1 Report
Good job!
Reviewer 2 Report
This paper describes that the "one-factor-at-a-time method" experiment was conducted, and regression analyses were used, which is not true. Actually, it is not true. Table 3 indicates that factional factorial design was employed. Please correct this part.
Figure 1 provides a good overview for the study, which is very effective to convey the research steps and ideas.
Added comments
1. Diamond Burnish process for hole surface integrity and sliding wear resistance are the major research question addressed by this paper.
2. Surface integrity is a machining characteristic that has been researched a lot in turning, milling, grinding and other machining processes, but relatively not common in diamond burnished holes. Plus this paper addressed sliding wear resistance in addition. The research methods, NSGA-II and finite analysis, are not completely new, but the application in this Burnishing process is significant.
3. The experiments and data collection are good. Maybe I did not catch it in the paper, how to obtain surface axis residual stress and hoop residual stress is not clear to me.
4. the conclusions are consistent with the evidence and arguments presented.
5. the references are appropriate
6. The researcher did a good job in presenting Figure 1. The comparison of Figure 11, 12, 13 and 14 are effective.
Reviewer 3 Report
The authors present a very well written, easy to follow, interesting work on the optimization of the diamond burnishing process. Some minor corrections are listed as well as some questions proposed.
-In page 1, line 13, please consider: This article outlines a technology for hole finishing in short-length cylinder lines production (…);
- All document: Please don’t use the comparative form of short (shorter) when the purpose of the sentence is not comparing hole length. Consider referring to the cylindrical holes as “short-length holes”;
-In page 2, line 100: Could you explain what “other parameters are correlated?”;
-In figure 2 please consider including (a) and (b) referring to the specimen and its drawing, respectively. Please update the caption accordingly;
-In line 241 the title seems unnecessarily long. Please consider changing to: Chemical composition, mechanical characteristics and initial microstructure;
-Please do not split table 2 in two pages.
-In line 385: How can the authors explain lower roughness for larger DB radius?
- Are the research contents and conclusions of the paper thoroughly compared with other research results?
-Could the authors include a scale in figures 15 and 16? How can the authors explain such strong transition between DB-ed and ground surfaces to worn surfaces? Should it not be a smoother transition?
-The authors mention strong pronunciation of carbide precipitates. How can this relate to the pits shown in figures 15 and 16.
-Have the authors considered the optical microscopy analysis of the burnishing tool? Even though no wear is expected, it would be interesting to show/confirm the integrity/sphericity of those tools.
